# AlphaFold prediction of structural ensembles of disordered proteins

Z. Faidon Brotzakis [1,2,3], Shengyu Zhang[1,3], Mhd Hussein Murtada [1,3] & Michele Vendruscolo [1] ✉

Deep learning methods of predicting protein structures have reached an accuracy comparable to that of high-resolution experimental methods. It is thus possible to generate accurate models of the native states of hundreds of millions of proteins. An open question, however, concerns whether these advances can be translated to disordered proteins, which should be represented as structural ensembles because of their heterogeneous and dynamical nature. To address this problem, we introduce the AlphaFold-Metainference method to use AlphaFold-derived distances as structural restraints in molecular dynamics simulations to construct structural ensembles of ordered and disordered proteins. The results obtained using AlphaFold-Metainference illustrate the possibility of making predictions of the conformational properties of disordered proteins using deep learning methods trained on the large structural databases available for folded proteins.

The application of deep learning methods to the protein folding problem has transformed our ability to generate accurate models of the native states of proteins from the knowledge of their amino acid sequences[1–5]. The initial predictions of the native structures of proteins have also been recently extended to protein complexes[5–7].

These advances have prompted the question of whether it is possible to use this type of approach for the prediction of the conformational fluctuations of the native states of folded proteins[8–16], and more generally for the characterisation of the structural properties of the native states of disordered proteins[17–20]. Support for this idea comes from the observation that AlphaFold performs as well as current state-of-the-art predictors of protein disorder[21,22]. It has also been reported that the predicted aligned error (PAE) maps from AlphaFold are correlated with the distance variation matrices from molecular dynamics simulations[9], suggesting that AlphaFold provides information about the dynamics of proteins in addition to their structures.

Since the native states of disordered proteins can be represented in terms of ensembles of conformations with statistical weights obeying the Boltzmann distribution[17,19,23,24], a relevant goal is to extend AlphaFold to predict structural ensembles. In this work, we propose an approach to perform this task. We base this approach on the observation that

AlphaFold can predict inter-residue distances even for disordered proteins, despite having been trained on folded proteins.

The possibility of performing predictions about disordered proteins based on the information available for ordered proteins is important. This is because it enables the transfer to disordered proteins of inter-residue distance information derived from folded proteins. For the training of deep learning methods, large numbers of high-resolution structures of folded proteins are available in the Protein Data Bank (PDB)[25]. By contrast structural ensembles of disordered proteins have been determined in much lower numbers and with less accuracy[23], making it challenging to use them to train deep learning methods. As an alternative, training on model structural ensembles derived from molecular simulations has been reported[26,27].

In the AlphaFold pipeline, the predicted inter-residue distances are provided in the form of a distance map (or distogram), from which the structure of a protein can be constructed[1]. Currently, however, structure predictions by AlphaFold for disordered proteins are not fully consistent with small-angle X-ray scattering (SAXS) data[18]. This is because, for a disordered protein, the prediction problem consists in the translation of the predicted distance map into a structural ensemble, rather than a single structure. There are many well-

[1]Centre for Misfolding Diseases, Yusuf Hamied Department of Chemistry, University of Cambridge, Cambridge, UK. [2]Institute for Bioinnovation, Biomedical Sciences Research Center "Alexander Fleming", 16672 Vari, Greece. [3]These authors contributed equally: Z. Faidon Brotzakis, Shengyu Zhang, Mhd Hussein Murtada. ✉e-mail: mv245@cam.ac.uk

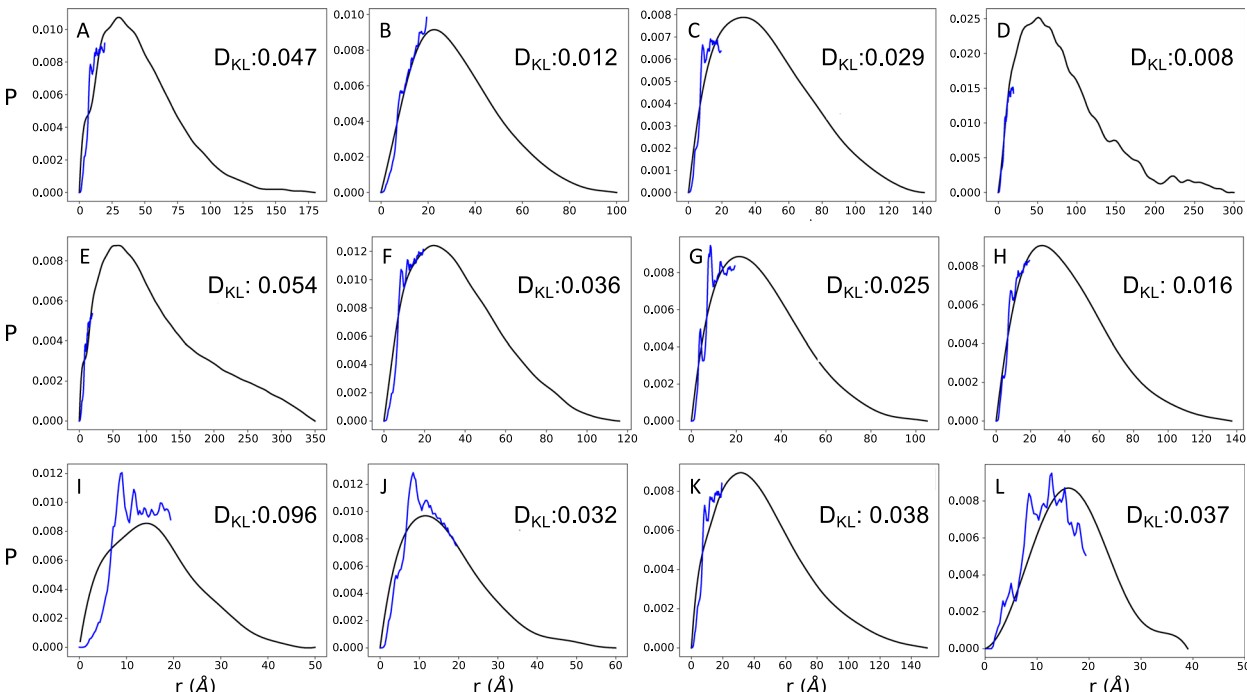

**Fig. 1 | Comparison of inter-residue distance distributions obtained by SAXS and predicted by AlphaFold for highly disordered proteins. A–K** Results for a set of 11 highly disordered proteins for which both SAXS and NMR diffusion measurements are available[36]. SAXS-derived inter-residue distance distributions are shown in black, and AlphaFold-predicted average inter-residue distance distributions are shown in blue. The cut-off distance in the AlphaFold predictions is 21.84 Å, so the blue lines stop at this value. For comparison, we report the Kullback–Leibler divergence ($D_{KL}$) between SAXS and AlphaFold-predicted average inter-residue distance distributions The proteins shown are: ANAC046 (**A**), A1 (**B**), ProTα (**C**), GHR-IDC (**D**), tau (**E**), Sic1 (**F**), DSS1 (**G**), NHE6cmdd (**H**), RS (**I**), Hst5 (**J**), and α-synuclein (**K**). **L** For comparison, we show the results for a folded protein (ubiquitin).

established ways to reconstruct a structural ensemble from a distance map.

Here we show that it is possible to use the predicted distances as structural restraints in molecular dynamics simulations, so that the resulting structural ensembles are consistent with the predicted distances[24,28,29]. Since our aim is to generate structural ensembles of proteins that contain disordered regions, we implemented the structural restraints according to the maximum entropy principle within the metainference approach[28]. We illustrate the resulting AlphaFold-Metainference approach for a set of well-studied highly disordered proteins[30] as well as for proteins that include both ordered and disordered domains, including TAR DNA-binding protein 43 (TDP-43), which is associated in amytrophic lateral sclerosis (ALS)[31], ataxin-3, which is linked with Machado–Joseph disease (also known as spinocerebellar ataxia type 3)[32], and the human prion protein[33], which causes Creutzfeldt–Jakob disease and related prion diseases[34].

## Results

### AlphaFold predicts accurate inter-residue distances

The AlphaFold-Metainference method that we report in this work is based on the observation that AlphaFold can predict the average values of inter-residue distances for disordered proteins (Fig. 1 and Supplementary Fig. 1). This feature may have not been evident so far likely because the reconstruction of individual structures from distance maps currently provided by AlphaFold is tailored toward the prediction of structured native states[1].

**SAXS-derived distance distributions.** Since obtaining experimental information about inter-residue distances for disordered proteins is challenging[24], comparing predicted and measured inter-residue distances requires some considerations. In techniques that exploit labels, such as fluorescence resonance energy transfer (FRET) and

paramagnetic relaxation enhancement (PRE) in nuclear magnetic resonance (NMR) spectroscopy, the presence of the labels themselves could affect the properties of the conformational ensembles[35–37]. Here, we used small-angle X-ray scattering (SAXS) data and NMR diffusion measurements, which offer label-free information about inter-residue distance distributions in disordered states of proteins. Our results show that there is a good agreement between the AlphaFold predictions of distance distributions and the SAXS-derived ones for a set of 11 proteins for which both SAXS measurements NMR diffusion measurements are available, and add a folded protein (ubiquitin, PDB:1UBQ [https://doi.org/10.2210/pdb1UBQ/pdb]) as control (Fig. 1 and Supplementary Fig. 2). Since AlphaFold predicts distances up to about 22 Å (see Methods), the AlphaFold-predicted distance distributions do not cover the entire SAXS-derived distributions. To obtain the SAXS-derived distance distributions from the SAXS profiles, we used a method described previously[38] (see Methods). The AlphaFold-derived distance distributions (distograms) are shown in Supplementary Fig. 3. We found a comparable value of $D_{KL}$ (0.037) for ubiquitin with respect to the 11 highly disordered proteins ($D_{KL}$ range: 0.008–0.096) (Fig. 1), further indicating that AlphaFold predicts inter-residue distances with comparable accuracy for ordered and disordered proteins.

**Distance maps derived from molecular simulations.** For further validation, we analysed recently reported structural ensembles of Aβ[39] and α-synuclein[40] obtained using all-atom molecular dynamics (MD) simulations, and coarse-grained simulations using CALVADOS-2 (C2), which are in good agreement with experimental data. The distances predicted by AlphaFold are in good agreement with those back-calculated from the MD ensembles of Aβ and α-synuclein (Supplementary Fig. 1A, B) and from the CALVADOS-2 ensembles (Supplementary Fig. 1C–M). Since AlphaFold predicts distances up to about 22 Å, this correlation stops at around this value.

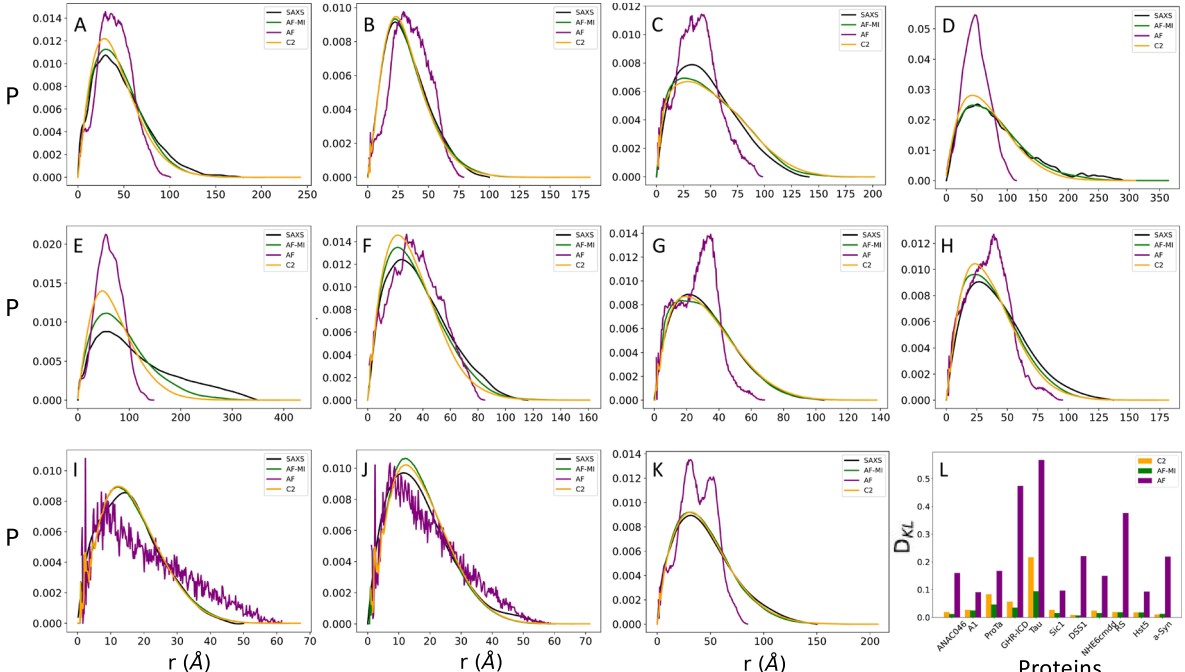

**Fig. 2 | Comparison of pairwise distance distributions for highly disordered proteins from SAXS data and from structural ensembles obtained by molecular simulations. A–K** Experimental pairwise distance distributions obtained by SAXS (SAXS, black lines) are compared with those directly calculated from the AlphaFold single-structure predictions (AF, purple lines), and the AlphaFold-Metainference structural ensembles (AF-MI, green lines). For comparison, the pairwise distance distributions obtained using CALVADOS-2 are also shown (C2, orange lines). The proteins are the same shown in Fig. 1A–K. **L** Quantitative assessment using the Kullback–Leibler divergence of the agreement between experimental and calculated distance probability distributions between SAXS and AlphaFold single-structure (purple), CALVADOS-2 (orange), and AlphaFold-Metainference (green).

## SAXS validation of highly disordered structural ensembles

As mentioned above, the use of SAXS measurements enables the calculation of pairwise distance distributions (Fig. 2A–K, black curves). We compared these experimentally-derived distance distributions with those obtained from structural ensembles determined from the AlphaFold-Metainference simulations (Fig. 2A–K, green curves) (see Methods) for the 11 highly disordered proteins described above. We chose this set of proteins because of the availability of SAXS data, and the range of length (24–414 residues) and scaling exponent ν (0.49–0.62) (see Methods and Supplementary Fig. 4). For comparison, we also show the distance distributions obtained using CALVADOS-2 (Fig. 2A–K, orange curves) and directly from the AlphaFold-derived distance distributions generated from individual AlphaFold structures (Fig. 2A–K, purple curves). To provide a quantitative comparison, we show that AlphaFold-Metainference and CALVADOS-2 provide structural ensembles in better agreement with SAXS data compared to individual AlphaFold-derived structures (Fig. 2L). Together with the comparison of the radius of gyration (Rg) values from AlphaFold-Metainference and AlphaFold with experimental SAXS values (Supplementary Fig. 4), these results indicate that individual AlphaFold structures of are not in good agreement with experimental SAXS data.

We further compared the structural ensembles using NMR chemical shifts, which were back-calculated at each time step using CamShift[41,42] (Supplementary Fig. 5). Although the structure-based predictions of chemical shifts can be made only with considerable errors[41,42] (Supplementary Fig. 5, grey bars), as an illustration we found that the HN chemical shifts for Sic1, from AlphaFold-Metainference are marginally more accurate than those from CALVADOS-2, while in all other cases we could not reliably rank the performances of the two approaches (Supplementary Fig. 5).

We further show how the highly disordered proteins described above span values for the scaling exponent ν that deviate from the Flory value for random coils (ν = 0.5) (Supplementary Fig. 4B). For these highly disordered proteins, AlphaFold-Metainference generates structural ensembles in better agreement with the Rg values derived from SAXS experiments. These results further illustrate how, when AlphaFold-predicted distances are applied as structural restraints in molecular simulations through the AlphaFold-Metainference approach, they generate accurate distance distributions (Fig. 2).

We also show the sequence separation of the AlphaFold-predicted distances that are used as restraints in AlphaFold-Metainference after introducing a filtering criterion described (see Methods, Supplementary Fig. 6). For these proteins, AlphaFold-Metainference tends to improve the agreement with the experimental SAXS data compared to CALVADOS-2, as measured using the Kullback–Leibler distance (see Methods) (Fig. 2L). This finding can be attributed to the introduction of short-range distances restraints (Supplementary Fig. 6).

## SAXS validation of partially disordered structural ensembles

To illustrate the application of AlphaFold-Metainference to partially disordered proteins, we considered a set of 6 proteins that contain both ordered and disordered domains, spanning a range sequence lengths, and for which SAXS data are available for validation (see Methods).

We first present the results for TDP-43, a multifunctional RNA-binding protein with a modular structure that allows it to engage in various cellular processes, including transcription, pre-mRNA splicing, and mRNA stability[31]. TDP-43 is also associated with ALS and other neurodegenerative diseases[31]. The sequence of TDP-43 comprises 414 amino acids, which form different domains (Fig. 3). These domains include a folded N-terminal domain (residues 1–76), a disordered region (residues 77–105), a folded RNA recognition motif (residues 106–176), a second disordered region (residues 177–190), another folded RNA recognition motif (residues 191–259), and an long disordered C-terminal domain (residues 274–414), which contains a glycine-rich region, is involved in protein-protein interactions, and harbors most of the mutations associated with ALS[43]. Because of the

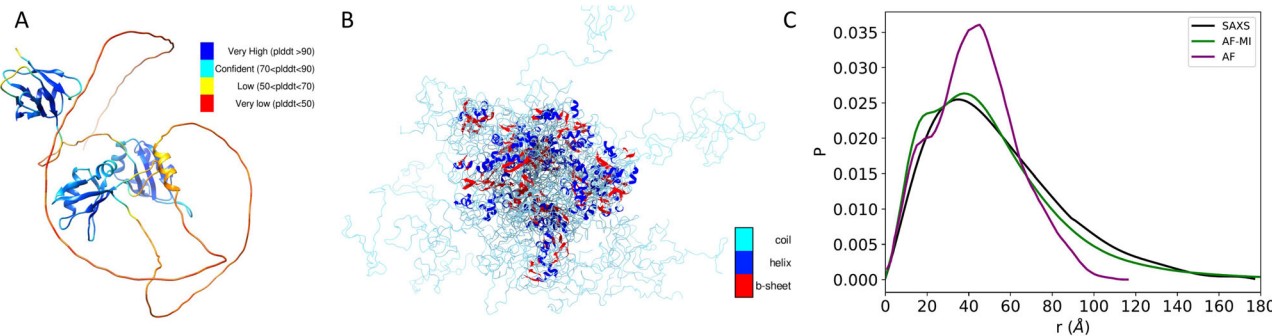

**Fig. 3 | Structural ensemble of TDP-43 predicted using AlphaFold-Metainference. A** An individual structure of TDP-43 predicted by AlphaFold[46]. The colors correspond to the predicted local distance difference test (pLDDT) score of AlphaFold, which is used to evaluate the accuracy of the predicted structure. The presence of long regions of low pLDDT score (red) indicates that the disordered

regions are not well predicted. **B** Structural ensemble of TDP-43 predicted by AlphaFold-Metainference. **C** Comparison between the pairwise distance distributions obtained by SAXS (black) with those predicted by AlphaFold single structure (purple) and by AlphaFold-Metainference (green).

presence of three disordered regions, the direct use of AlphaFold from the AlphaFold Protein Structure Database[44] results in a predicted structure spanning high accuracy regions with high Predicted Local distance difference test (pLDDT) confidence score at the N-terminal domain and RNA recognition motifs (pLDDT > 90) and low pLDDT confidence score for disordered regions (pLDDT < 50) (Fig. 3A) leading to an overall low agreement with the SAXS data (Fig. 3C and Fig. 6D), with a high $D_{KL}$ value (0.582). However, when applied our filtering criteria (see Methods) to select AlphaFold-predicted distances, followed by applying AlphaFold-Metainference with these distance restraints, we obtained a structural ensemble (Fig. 3B), which is in better agreement with the SAXS data, as measured by a $D_{KL}$ value of 0.018 (Fig. 3C and Fig. 6D).

To explore further the possibility of using AlphaFold-Metainference to predict structural ensembles of partially disordered proteins, we then analysed ataxin-3. The structure of this protein consists of an N-terminal Josephin domain (residues 1–182), and a long, disordered C-terminal domain that hosts the poly-Q tract[32] (Fig. 4). Again, because of the presence of long disordered regions, the use of AlphaFold from the AlphaFold Protein Structure Database resulted in a predicted structure (Fig. 4A) not in good agreement with the SAXS data (Fig. 4C and Fig. 6D), with $D_{KL}$ value of 0.653. However, when applied our filtering criteria to select AlphaFold-predicted distances for the AlphaFold-Metainference simulations, we obtained a structural ensemble (Fig. 4B) in better agreement with the SAXS data, as quantified by a $D_{KL}$ value of 0.020 (Fig. 4C and Fig. 6D).

Next, we report the results that we obtained using AlphaFold-Metainference for human prion protein, which has been intensively studied because of its role in Creutzfeldt–Jakob disease and related prion diseases[34]. This protein comprises a highly disordered N-terminal region that contains a series of octapeptide repeats involved in binding metal ions, and a folded C-terminal region that consists of three α-helices and two short β-strands (Fig. 5). The long, disordered N-terminal region makes it challenging to use AlphaFold directly from the AlphaFold Protein Structure Database. The structure predicted in this way (Fig. 5A) is not in good agreement with the SAXS data (Fig. 5C), with a $D_{KL}$ value of 0.1. However, when applied our filtering criteria, we obtained a structural ensemble (Fig. 5B) in better agreement with the SAXS data, as quantified by a $D_{KL}$ value of 0.053 (Fig. 5C and Fig. 6D).

Other 3 proteins (CbpD, H16, and PC) that we studied are shown in Fig. 6A–C. Overall, these results show that in all cases the agreement between experimental and back-calculated inter-residue distance distributions is very good, and much improved with respect to AlphaFold individual structures from the AlphaFold Protein Structure Database (Fig. 6D).

To compare the structural ensembles generated by using AlphaFold-predicted distances as structural restraints within the AlphaFold-Metainference approach and those generated using CAL-VADOS-2, where only the folded domains were restrained (RMSD-C2), we calculated the respective $D_{KL}$ values (Fig. 6D). We found that for four (ataxin-3, CbpD, H16, and PC) of the six proteins analyzed, AlphaFold-Metainference performs better than RMSD-C2, while for the remaining two (TDP-43 and human prion protein) the two methods produce comparable structural ensembles, a result that can be attributed by the relatively low number of long-range restraints from the AlphaFold predictions (Supplementary Fig. 6).

## Discussion

We described the AlphaFold-Metainference method of generating structural ensembles representing the native states of disordered proteins and of proteins containing disordered regions. The method is based on the observation that the inter-residue distances predicted by AlphaFold are relatively accurate even for disordered proteins (Figs. 1 and 2 and Supplementary Fig. 1), so that they can be used as structural restraints in molecular dynamics simulations within the metainference framework (Figs. 3–6).

The finding that AlphaFold can predict inter-residue contacts for disordered proteins, despite the absence of disordered proteins in the AlphaFold training dataset, may perhaps seem surprising. Since deep learning methods are known for not being able to readily generalise to cases not encountered during training, this result may indicate that the types of interactions between residues that stabilise the native states of disordered proteins do not fundamentally differ from those that stabilise the native states of folded proteins, but they may just be collectively less stable. This possibility is also consistent with previous studies that reported that proteins in unfolded states sample preferentially intermolecular contacts present also in their native states[45–49].

To generate structural ensembles, AlphaFold-Metainference translates the AlphaFold distograms into structural ensembles using the predicted distances as structural restraints in molecular dynamics simulations[24,28,30]. We also showed previously that the predicted distance maps can be employed in a reweighting approach, where one can use as starting point structural ensembles obtained by molecular dynamics simulations[50].

Overall, the results that we presented illustrate the use of deep learning methods originally developed for predicting the native states of folded proteins to generate structural ensembles representing the native states of disordered proteins. The scope of protein structure predictions based on deep learning can thus be considerably extended. We note that the use of all-atom force fields could increase the accuracy of the resulting structural ensembles, although at the price of longer

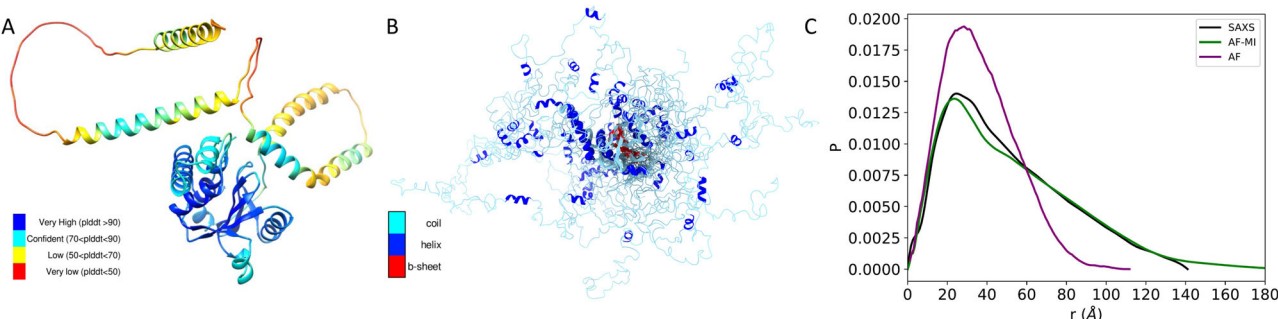

**Fig. 4 | Structural ensemble of ataxin-3 predicted using AlphaFold-Metainference. A** An individual structure of ataxin-3 predicted by AlphaFold[46]. The colors correspond to the predicted local distance difference test (pLDDT) score of AlphaFold. The presence of long regions of low pLDDT score (red) indicates that the disordered regions are not well predicted. **B** Structural ensemble of ataxin-3 predicted by AlphaFold-Metainference. **C** Comparison between the pairwise distance distributions obtained by SAXS (black) with those predicted by AlphaFold single structure (purple) and by AlphaFold-Metainference (green).

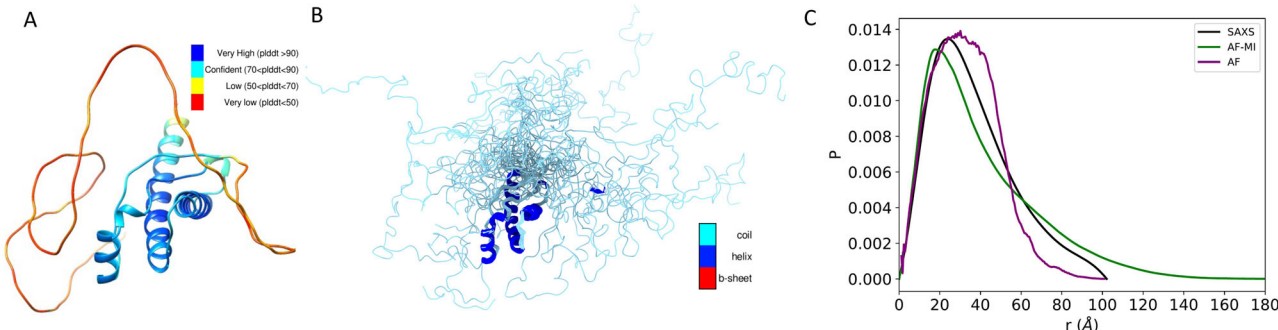

**Fig. 5 | Structural ensemble of human prion protein predicted using AlphaFold-Metainference. A** An individual structure predicted by AlphaFold[46]. The colors correspond to the predicted local distance difference test (pLDDT) score of AlphaFold. The presence of long regions of low pLDDT score (red) indicates that the disordered regions are not well predicted. **B** Structural ensemble predicted by AlphaFold-Metainference. **C** Comparison between the pairwise distance distributions obtained by SAXS (black) with those predicted by AlphaFold single structure (purple) and by AlphaFold-Metainference (green).

simulation times, and that combining AlphaFold-predicted distances with experimental data, including NMR spectroscopy, SAXS, and cryo-electron microscopy (cryo-EM), would also be possible using the metainference framework.

## Methods

### AlphaFold-predicted distances

Average inter-residue distances were predicted through the distogram head of AlphaFold[1]. These distances are defined between the β carbon atom positions for all amino acid types except glycine, for which the α carbon atom positions were instead used. The multiple sequence alignment (MSA) was conducted by MMseqs2[51] (default setting) on BFD/MGnify[3] and Uniclust30[52]. Model 1.1.1 of AlphaFold (default setting)[1] was used for the predictions, with no structural templates. AlphaFold provides distributions of inter-residue distances into 64 bins of equal width, covering the range from 2.15625 to 21.84375 Å, with the last bin also including distances longer than 21.84375 Å. For each pair of residues ($i$ and $j$), AlphaFold predicts the probability $p_{ij}^b$ that their distance is within bin $b$. The predicted distance $\hat{d}_{ij}$ and the standard deviation $\sigma_{ij}$ of the predicted distribution of the distances between residue $i$ and $j$ are calculated by

$$\hat{d}_{ij} = \sum_{b=1}^{64} d^b p_{ij}^b \tag{1}$$

$$\sigma_{ij} = \sqrt{\sum_{b=1}^{64} \left( d^b - \hat{d}_{ij} \right)^2 p_{ij}^b} \tag{2}$$

where $d^b$ represents the central value of bin $b$.

### Metainference

Metainference is a Bayesian inference method that enables the determination of structural ensembles by combining prior information and experimental data according to the maximum entropy principle[28]. In this work, we implemented this method by using the distogram (or distance map) $d^{AF}$ predicted by AlphaFold as pseudo-experimental data. By design, metainference can disentangle structural heterogeneity from systematic errors, such as force field or forward model inaccuracies, random errors in the data, and errors due to the limited sample size of the ensemble[28]. The molecular simulations are carried out according to the metainference energy function, $E = -k_B T \log(p_{\mathrm{MI}})$, where $k_B$ is the Boltzmann constant, $T$ is the temperature, and $p_{\mathrm{MI}}$ is the metainference, maximum-entropy-compatible, posterior probability

$$p_{\mathrm{MI}}(\mathbf{X}, \sigma^{SEM}, \sigma^B | \mathbf{D}) = \prod_{r=1}^{N_R} \mathrm{p}(X_r) \prod_{i=1}^{N_D} \mathrm{p}(\mathbf{D} | \mathbf{X}, \sigma_i^{SEM}, \sigma_{r,i}^B) p(\sigma_{r,i}) \tag{3}$$

In this formula, $\mathbf{X}$ denotes the vector comprising the atomic coordinates of the structural ensemble, consisting of individual replicas $X_r$ ($N_R$ in total), $\sigma^{SEM}$ the error associated to the limited number of replicas in the ensemble, $\sigma_B$ the random and systematic errors in the prior molecular dynamics force field as well as in the forward model and the data, and $d^{AF}$ the AlphaFold distogram. Note that $\sigma^{SEM}$ is calculated for each data point ($\sigma_i^{SEM}$), while $\sigma^B$ is computed for each data point $i$ and replica $r$ as $\sigma_{r,i}^{B}$. The functional form of the likelihood

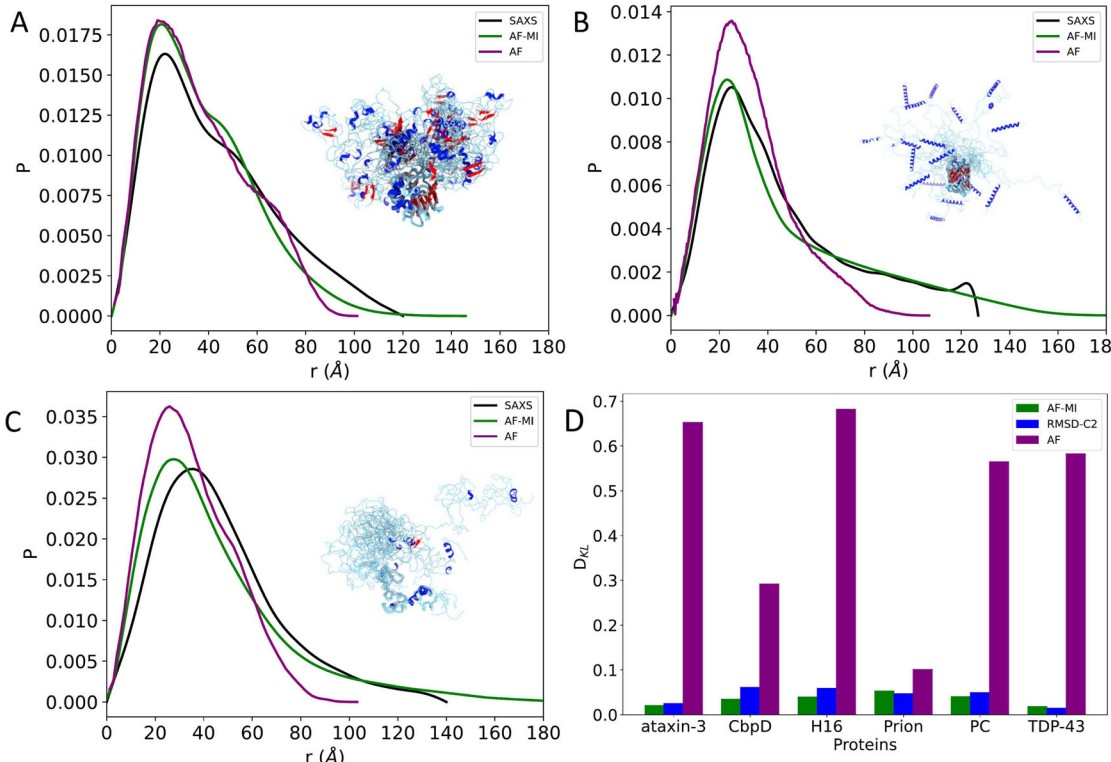

**Fig. 6 | Comparison of SAXS-derived and AlphaFold-predicted pairwise distance distributions for partially disordered proteins. A–C** Experimental pairwise distance distributions obtained by SAXS are shown in black, by AlphaFold individual structures (AF) in purple, and by AlphaFold-Metainference structural ensembles (AF-MI) in green. The proteins shown are CbpD (A), H16 (**B**), and PC (**C**).

**D** Comparison of the Kullback–Leibler distances between AlphaFold individual structures (AF) and SAXS (purple), CALVADOS-2 with RMSD restraints (RMSD-C2) and SAXS (blue), and AlphaFold-Metainference (AF-MI) and SAXS (green) for the 6 partially disordered proteins shown in Figs. 3–6.

$p(\boldsymbol{d}^{AF} \mid \mathbf{X}, \sigma_i^{\text{SEM}}, \sigma_{r,i}^{\text{B}})$ is a Gaussian function

$$p\left(\boldsymbol{d}^{AF} \mid \mathbf{X}, \sigma_i^{SEM}, \sigma_r^B\right) = \frac{1}{\sqrt{2\pi}\sqrt{(\sigma_{r,i}^B)^2 + (\sigma_i^{SEM})^2}} \exp\left[-\frac{1}{2}\frac{d_{i,j}^{AF} - d_{ij}(\mathbf{X})^2}{(\sigma_{r,i}^B)^2 + (\sigma_i^{SEM})^2}\right] \quad (4)$$

where $d_{i,j}(X)$ represents the forward model for data point $i,j$, namely the $i,j$ distance calculated in the structural ensemble. For multiple replicas, the metainference energy function is

$$E_{MI}(\boldsymbol{X}, \sigma) = E_{MD}(\boldsymbol{X}) + \frac{k_B T}{2} \sum_{r,i}^{N_R, N_D} \frac{[d_i - f_i(X_r)]^2}{(\sigma_{r,i}^B)^2 + (\sigma_i^{SEM})^2} + E_\sigma \quad (5)$$

where $E_\sigma$ corresponds to the energy term associated with the errors

$$E_\sigma = k_B T \sum_{r,i}^{N_R, N_D} -\log(\sigma_{r,i}^B) + \frac{1}{2}\log\left[(\sigma_{r,i}^B)^2 + (\sigma_i^{SEM})^2\right] \quad (6)$$

Finally, $E_{MD}$ corresponds to the potential energy function of the molecular dynamics force field, which in this case is the CALVADOS-2 force field[53]. While the space of conformations $X_r$ is sampled through multi-replica simulations (in this study we used six replicas) the error parameters for each datapoint $\sigma_{r,i}^{\text{B}}$ are sampled through a Gibbs sampling scheme at each time step[28]. The range of the error sampling was [0.0001,10] and the associated trial move error perturbation of the Gibbs sampling was 0.1. The error parameter due to the limited number of replicas used to estimate the forward model ($\sigma^{\text{SEM}}$) was calculated on the fly by window averaging every 200 steps of molecular dynamics.

## Selection of AlphaFold-predicted distance restraints
The AlphaFold-predicted distance maps corresponding to the proteins presented in the main text are shown in Supplementary Fig. 3. Since AlphaFold is not trained to predict distances above 21.84 Å, we excluded distances with $p_{i,j}\left(r \geq 21.84\,\text{Å}\right) > 0.02$. For the selected distances, there is a good correlation between the AlphaFold-predicted distances with those back-calculated from molecular dynamics simulations and from CALVADOS-2 (Supplementary Fig. 1). In addition, we further selected the distances using the predicted alignment error (PAE) of AlphaFold, which provides a metric of accuracy for predicted distances (Supplementary Fig. 8). For this purpose, we used as benchmarking comparison a subset of 6 proteins for which CALVADOS-2 results in relatively large deviations from the experimental Rg values[54] (Supplementary Fig. 7). This approach led us to define the following distance selection criterion. For hydrophilic proteins (as measured by a Kyte−Doolittle hydropathy score < 1.4), which tend to be more disordered, with at least a 5-residue stretch with a predicted local distance difference test (pLDDT) score > 75, AlphaFold-predicted distances with PAE < 10 are selected. Otherwise, distances with PAE < 5 are selected. Furthermore, as the CALVADOS-2 coarse-grained model used in the AlphaFold-Metainference simulations does not maintain the secondary structure, we employed the pLDDT score to select the inter-residue distances corresponding to structured regions. Sequence regions of at least two consecutive residues with a pLDDT score > 0.75 were considered structured regions, and restrained to the AlphaFold-predicted structure by using an upper root mean square distance (RMSD) wall. The residue distances corresponding to these structured regions were excluded from the distance restraints set. This selection may not be necessary if one uses all-atom force fields that are able to maintain the protein secondary structure.

## Generation of the structural ensembles

To sample efficiently the conformational space of disordered proteins using average inter-residue distances predicted by AlphaFold as structural restraints, in this work we implemented AlphaFold-Metainference with coarse-grained simulations within the metainference framework (the use of all-atom simulations is also possible, although more computationally expensive). To carry out this procedure, we implemented the CALVADOS-2 force field[53] into OpenMM[35], thereby enabling the use of PLUMED[56], where corresponding code is available in GitHub (https://github.com/vendruscolo-lab/OpenMM-Plumed-MPI). After generating the CALVADOS-2 force field for each protein sequence by using a previously reported procedure[53], we performed a short energy minimization of 100 steps. Then, all molecular dynamics simulations started from the structures predicted by AlphaFold. The simulations were performed in the NVT ensemble with $10^6$ steps per replica (six replicas in this study), starting with different initial positions obtained from the energy minimization step, with parameters (temperature, pH and ionic strength) reported in Table S1. We used the Langevin integrator with a time step of 5 fs and friction coefficient of 0.01 ps$^{-1}$. The Cα-based model was implemented using the CALVADOS 2 parameters and functional forms[53].

For the highly disordered proteins and partially disordered proteins considered in this work, we then used PULCHRA[57] to generate all atom representations of all the structures in the coarse-grained ensembles, followed by an energy minimization using GROMACS[58]. PULCHRA is a fast and robust method for the reconstruction of full-atom structures that starts from a reduced protein representations such as CA atoms and applies molecular a mechanics force field and a subsequent backbone and side chain optimization to generate full atom representations[47]. This procedure is available at https://github.com/vendruscolo-lab/AlphaFold-IDP/blob/main/backmap/backmap_abeta_c2/script.sh.

We implemented the parallel bias metadynamics[59] the since the timespan of conformational transitions of disordered proteins exceed the μs timescale. The biasing collective variables (CVs) and metadynamics parameters are reported in Table S1. To generate the final unbiased structural ensembles while taking into consideration the parallel bias metadynamics weights, we adopted a previous protocol[60,61]. First, the replicas (six per simulation in this study) were concatenated into a single trajectory followed by generation of the final metadynamics bias per frame by increasing the bias deposition pace. Secondly, we generated the Torrie Valleau weights for each frame of the single trajectory employing the per-frame-bias value. The final structural ensembles were generated by resampling the concatenated trajectory by these Torrie Valleau weights. For the convergence analysis, we divided the per-replica trajectories into five segments of increasing coverage and used the respective Torrie Valleau weights to plot time five time-dependent free energy projection estimates along the biased CVs (Supplementary Fig. 9), after removing the first 10% of the trajectory per-replica as equilibration. The combination of metadynamics with the CALVADOS-2 coarse-grained model enables the rapid convergence of the structural ensembles.

## Pairwise distance distribution functions

The pairwise distance distribution functions in Figs. 2–6 were constructed for AlphaFold-Metainference, CALVADOS-2 and AlphaFold as follows. For AlphaFold-Metainference and CALVADOS-2, we calculated for each conformation in the structural ensemble the inter-residue distances and binned them in a normalized histogram, respectively into the green curves (AF-MI) or the orange curves (CALVADOS-2). For the individual structures from the AlphaFold Protein Structure Database (purple curves), we calculated the inter-residue distances and binned them in a normalized histogram.

## Comparison of SAXS distributions

The agreement between SAXS-derived ($P(r)$) and simulated ($Q(r)$) pairwise distance distributions were calculated using the Kullback−Leibler distance

$$D_{KL}(P(r), |, Q(r)) = \sum_{i=1}^{N} P(r_i) \log \frac{P(r_i)}{Q(r_i)} \qquad (7)$$

## Proteins simulated with AlphaFold-Metainference

For the study of highly disordered proteins, we used a previously reported set of 11 proteins for which SAXS measurements are available (Fig. 1A–L). For the study of partially disordered proteins, we used a set of 6 proteins of varying sequence length (L), radius of gyration (Rg), and for which SAXS measurements are available (Figs. 3–6): TDP-43[62], an ataxin-3 variant containing a 16-residue-long poly-Q tract (ataxin-3, SASDJ47)[32], human prion protein (prion, SASDNB8 [https://www.sasbdb.org/data/SASDNB8/])[33], chitin-binding protein D (CbpD, SASDK42)[63], exon1 of a non-pathogenic form of huntingtin containing a 16-residue-long poly-Q tract (H16, SASDQR8)[64], and human vitamin K-dependent protein C (PC, SASDJC6)[65]. The SAXS profiles for these proteins show a characteristic partially disordered protein profile, that is a combination of a bell-shape and a plateau that slowly decays to zero (Figs. 3–6).

## Reporting summary

Further information on research design is available in the Nature Portfolio Reporting Summary linked to this article.

## Data availability

All the data from the MD simulations, including the coordinate file of the final output for each MD simulation, as well as the PLUMED input, output and analysis files, are available at https://doi.org/10.5281/zenodo.14712644. SASBDB codes of previously published small angle scattering data used in this study are SASDJ47, SASDNB8, SASDK42, SASDQR8 and SASDJC6. The PDB code of the previously published structure used in this study is 1UBQ. Source Data are provided as a Source Data file. Source data are provided with this paper.

## Code availability

The code to prepare an AlphaFold-Metainference simulation can be found at https://github.com/vendruscolo-lab/AlphaFold-IDP. A tutorial for running AlphaFold-Metainference simulations is available at https://github.com/vendruscolo-lab/AlphaFold-MetaInference-Tutorial/tree/main and https://www.plumed-tutorials.org/lessons/24/014/data/NAVIGATION.html[66].

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

## Acknowledgements

We would like to acknowledge funding from UKRI (10059436 and 10061100). Z.F.B. would like to acknowledge funding from the Horizon Europe Programme under the Widening Participation & Spreading Excellence component (call Twinning HORIZON-WIDERA-2022-TALENTS-01-01–ERA Chairs), Project "Boost4Bio" (Grant Agreement No. 101087471), and the Bodossaki Foundation postdoctoral research fellowship.

## Author contributions

Z.F.B., S.Z., and M.V. conceived the project. Z.F.B, S.Z., and M.H.M. performed experiments. All authors analyzed data and wrote the article.

## Competing interests

The authors declare no competing interests.
