## [Transparent Peer Review file · Nature Communications]

AlphaFold Prediction of Structural Ensembles of Disordered Proteins

Corresponding Author: Professor Michele Vendruscolo

Version 0:

Reviewer comments:

Reviewer #1

(Remarks to the Author)

Brotzakis, Zhang, and Vendruscolo present a very important finding that AlphaFold captures the inter-residues distances of disordered proteins. This is an unexpected finding as AlphaFold was famously trained on folded proteins only. The authors suggest folded and unfolded regions only differ in stability but are otherwise the same and thus AlphaFold is able -in principle - to provide important data on conformational ensembles of disordered regions. They further show how this information can be used to arrive at conformational ensembles of disordered proteins. Rather ingeniously they use the distance map which AlphaFold is using to generate structures to reweight pre-calculated conformational ensembles employing well-established Bayesian/maximum entropy methods. Overall, this manuscript represents an important advance.

1) My main suggestion is to provide, e.g., Jupyter notebooks of at least one of the example cases. This will help others to use this approach and develop further improvements which for sure will strengthen the impact of this work further. It will be important to illustrate in such an example how the distance map is extracted from AlphaFold.

2) In Fig S4-S8 absolute NMR chemical shifts are compared for experiments and computational models. I think it would be better to plot secondary chemical shifts for comparisons so that deviations between models and experiments can be assessed, which is typically what is done for such comparison. In case their CamShift chemical shift prediction method warrants a different approach it would be good to make this clear in the manuscript.

3) For α -synuclein an RG value of 36 Å was determined by Araki et al Scientific Reports 2016. This value is often thought as the best experimental estimate as to RG value of α -synuclein. It looks like the reweighted CALVADOS2 ensemble is close to this value. This maybe partially explained by the composition of the CALVADOS2 training set, but it may still be worth mentioning this encouraging agreement with experiments?

4) For the SAXS comparison in Fig 6 it may be good to compare experimental SAXS profiles with a back calculated SAXS profiles at least for one of the proteins investigated here. The caption misses an explanation for panel H.

5) The group of M. Blackledge has studied partially folded α -helices in disordered proteins by NMR. Would the presented approach also be applicable to such cases?

6) To demonstrate the value of the distance map from AlphaFold, the authors may consider running a restrained (replica averaged) molecular dynamics simulation for a tractable model system. I suspect that such restraints could greatly improve atomistic models of disordered peptides, which could strengthen the key conclusions of this paper.

Minor points

1) For the correlation plots in Figure 2 it would help if the length of the axis is the same (i.e., the plot is square) so that one can compare more clearly MD and AlphaFold

2) Several methods have been proposed to sample different conformational states with AlphaFold. Maybe the authors could comment briefly on whether methods trying to sample conformational diversity may be required for their proposed AlphaFold approach to generate IDP conformational ensembles?

3) Could the authors provide some more details as how they energy minimised the models with GROMACS? I.e., which force field was used? I understand that this is a minor point, but including this information may be useful for others who want to build on the presented approach.

Reviewer #2

(Remarks to the Author)

In this manuscript Brotzakis et al., compare AlphaFold distance predictions with experimental SAXS distance distributions, as well as show how AlphaFold distance predictions can be utilized to reweight different types of conformational ensembles.

The current version of this manuscript lacks significance to the field, as the authors make general statements that do not reflect the details of the data shown. If the authors can provide additional analyses to show under what conditions AlphaFold predictions are helpful, then the manuscript would be significant to the field.

More specifically, the authors claim that distances predicted by AlphaFold for disordered proteins are accurate but this statement is too general as the data reported shows caveats to this statement which the authors do not fully address in the text. For example, there are cases where unrestrained coarse-grained models do a better job than coarse-grained models that take into account AlphaFold distance constraints. Thus, it is currently unclear under what conditions one should use AlphaFold constraints versus when these constraints are not helpful. Overall the authors need to provide some additional analyses to show when (which distances and IDR properties) the distances predicted by AlphaFold are accurate as addressed in the points below.

On page 3, the authors state, "Our results show that there is a good agreement between the AlphaFold predictions of distance distribution functions and the SAXS-derived ones for a set of 7 proteins for which both SAXS measurements NMR diffusion measurements were available." The authors use 7 of the initial set of 11 IDPs from the paper of Pesce et al. The authors must explain why these 7 were chosen rather than all 11. Additionally, the authors should provide information about these 7 IDPs including information such as their sequence, sequence length, scaling exponents, etc. This is important since the authors are claiming the AlphaFold prediction shows good agreement but it is not clear whether these IDPs chosen are representative and thus whether this result would hold for any IDP without the additional information about relative representation of the IDPs chosen.

Particularly, the authors do not compare the AlphaFold predictions of A1 which is the most compact IDP of the Pesce et al., set with the only scaling exponent below 0.5. Thus, it is not clear if A1 was left out because it does not show good agreement or some other reason, and this should be remedied. Overall, unless there is a feasibility issue for the other 4 proteins, the comparison between their SAXS-derived distributions and the AlphaFold predictions should be shown. Even if the authors do not include these 4 IDPs in Figure 6, the additional 4 IDPs need to be included in Figure 1. Without including A1 in particular, it will be unclear whether AlphaFold predictions hold for more compact IDPs.

Additionally, the authors should clarify whether just the disordered regions (i.e., the regions used by Pesce et al.,) or the full protein sequences were used when generating the AlphaFold distograms.

On page 3, the authors state, "Since AlphaFold predicts distances up to about 22 Å (see Methods), the AlphaFold distribution does not cover the entire SAXS-derived distribution." Since the AlphaFold prediction includes a cutoff it would be helpful to know the distribution of $|j-i|$ lengths that are generally included in these distances, where i and j denote the residue position along the sequence of residue i and j . For instance, are most of the AlphaFold predictions accounting for local conformational information, i.e., within ~ 10 residues along the sequence from each other? How often is long range information ($|j-i| > \sim 20$) observed?

Furthermore, according to Wallner et al., (<https://doi.org/10.1101/2022.12.08.519560>) there is relatively good agreement between the distogram distances and the PDB model distances. Thus, it would be helpful to show whether using the PDB model distances using the AlphaFold cutoff of ~ 22 also show good agreement with the SAXS-derived distributions, i.e., how does imposing a single structure with no probability of distances compare to the SAXS-derived distributions? If the PDB model breaks down it would better highlight the importance of using the distogram. If the PDB alone is reasonable, then it would be interesting to know at what length scale the PDB model breaks down since distances greater than ~ 22 can be included.

On page 3, the authors state, "The correlation reported in Figure 2 indicates that AlphaFold can predict inter-residue distances in the disordered native states of Ab and a-synuclein up to about 16 Å, which is about 70% of the cut-off distance in the AlphaFold predictions (see Methods)." How is a cutoff of 16 chosen? A correlation or R^2 should be calculated using different cutoffs to show the validity of this statement.

Also what is the average $|j-i|$ that such a cutoff is accounting for? Is this mostly local interactions, i.e., distances between residues close in sequence space. The authors should show if the data is split into distances where $|j-i|$ is less than or equal to ~ 10 and if $|j-i| > \sim 10$ (or a full titration of $|j-i| < x$) whether the distances that correspond to residues close in sequence space have a higher correlation than the distances that correspond to residues farther apart in sequence space. This would also help to know what kind of interactions AlphaFold gets right. Is it just distances between residues close in sequence space or is the variability not restricted to a particular $|j-i|$ range?

Furthermore, AlphaFold predicted distances are implied to be accurate based on the comparison to the MD extracted distances, yet no comparison of the MD simulations to experimental data is provided at this point in the manuscript (besides a citation) to show that the MD simulations are a good test of accuracy. At least comparing the α -synuclein distance distributions to the SAXS data, since this is being used as the gold standard in this manuscript, should be provided.

The representative conformations shown in Figures 3B, 3E, 4B, 5B, and 5F need better contrast since they are hard to see. The authors also need to explain how the representative conformations were picked. It is also not clear what the inclusion of these representative conformations provide.

On page 5, the authors state, "These results are consistent with previous findings that reweighting with many constraints produces posterior ensembles of high quality even from prior ensembles of relatively quality." There appears to be a word missing before the last "quality".

For A β , the C2 and AF-C2 (or just AF) distances are often very different, yet both ensembles show equally good agreement with the NMR data (Figure S6G). The authors should explain what this means, especially in the context of the fact that there are no experimental distances to compare to so which model is correct is unclear. Thus, if both do an equally good agreement with the NMR data is there a time when people should just use C2 vs AF-C2? Also in the main text the authors refer to Figure S6F when they seem to mean S6G.

On page 7, the authors state, "The AF-MD ensemble shows marginal improvement with respect to the MD ensemble in the chemical shifts of all types of atoms (Figure S7F)." It is unclear why the authors claim "marginal improvement...in the chemical shifts of all types of atoms" when only cb and maybe hn show a lower RMSD in AF-MD compared to MD. Thus, this statement needs to be updated to reflect the data.

On page 8, the authors state, "The use of the AlphaFold-predicted pairwise distances as constraints in the reweighting procedure tended to slightly improve the agreement with the experimental data, as measured using the Kullback-Leibler distance (see Methods) between distributions (Figure 6K)". First, the authors appear to be referring to 6H not 6K. Second, only ProTa and GHR-ICD have smaller Kullback-Leibler distance given the error bars, thus it is unclear why the authors claim AlphaFold-predicted constraints generally improve agreement. Thus, this statement is incorrect and must be changed to reflect the data. Third, the figure caption of Figure 6 does not match the subplots. Although it could be assumed panels A-G refer to the proteins in the order they are listed in H this should be clarified on the plot or in the figure caption. Additionally, the authors must also clarify what the error bars mean in panel H.

Lastly, the authors spend a decent amount of the paper showing how α -synuclein ensembles can be reweighted to match AlphaFold-predicted distances only to show that the C2 model without reweights is a better approximation of the experimental data. Thus, it is unclear when / why reweighting to AlphaFold-predicted distances would be helpful. Given that some IDPs do better using the AlphaFold-predicted constraints and some IDPs do worse / the same, the authors should provide some sort of analysis to try to explain this and thus give the readers a sense of when using the AlphaFold-predicted constraints would be more beneficial than just using the C2 model.

Additionally, again the authors should explain why only 7 of the 11 Pesce et al., proteins were chosen to examine in Figure 6. Unless there is a feasibility issue, the remaining 4 proteins should also be included here.

In the abstract, the authors statement "inter-residue distances predicted by AlphaFold for disordered proteins are accurate". This statement needs to be toned down and caveats need to be included.

On page 10, the authors state, "In practice, rather than constraining all distances, we did not include neighbouring residues (up to 2 residues) for A β and that have a distance larger than 21.8 Å in the MD or FD ensembles. For the α -synuclein calculations, we did not include neighbouring residues (up to 2 residues) and that have a distance larger than 21.8 Å in the MD ensemble." Why are these constraints only used for some of the ensembles? i.e., were these same constraints not used for the C2 ensembles?

A general issue that must be addressed is the use of comparison to SAXS derived distance distributions to show that AlphaFold is accurate (Figure 1). Given the data shown in Figure 6, it appears that a coarse-grained model with no constraints is often comparable to the AlphaFold distance probabilities up to ~20 Å. Thus, the question becomes are all models equally good at comparing to SAXS derived distances distributions up to this cutoff? If all models are equally good, then how does that imply AlphaFold is accurate / why should one use AlphaFold distances? I suggest that the authors provide a Kullback-Leibler distance measure for the single structure PDB from AlphaFold, the AlphaFold distogram distances, and the C2 model up to 22 Å. At least some comparison of accuracy should be included since the authors want to make the claim that using AlphaFold distances are helpful.

On page 8, the authors state, "Overall, the results that we have presented illustrate the use of deep learning methods originally developed for predicting the native states of folded proteins¹⁻³ to generate structural ensembles representing the native states of disordered proteins, or of proteins containing disordered regions." This statement is too general and must be changed to reflect what the data shows. The AlphaFold predictions are not always beneficial. Thus, the authors need to provide analyses to show when the AlphaFold predictions are beneficial to avoid misleading claims.

Reviewer comments:

Reviewer #2

(Remarks to the Author)

Overall the new results and text provided for the partially disordered proteins is clear. However, the authors should button up the text on the disordered proteins and show a few more analyses / controls in order to clean up the full story. The general takeaway the authors appear to describe is that using the AlphaFold structure directly from the web server is not particularly good at predicting the conformational ensemble. However, when AlphaFold-predicted average distances are used as restraints for disordered regions in simulations, then conformational ensemble information is in good agreement to SAXS.

To support this takeaway and the need for using AlphaFold restraints the authors should at least provide the following:

1. Histograms predicted by AlphaFold for each of the disordered proteins in the supplementary. These plots will show whether AlphaFold predicted distances ever show long range interactions or whether the restraints are generally just on local interactions. Additionally, the authors need to say whether these restraints are mainly on local interactions or not in the main text. This could also be provided by supplementary figures that show the average sequence separation or distribution of sequence separations used for the restraints.
2. A more detailed description of Figure 2 in the main text (see point 8 below). Currently the main text does not mention what the models for each of the curves are. This description should include a comparison between CALVADOS-2 results with and without restraints. One of the paper's main claims is that when AlphaFold-predicted average distances are used as restraints in simulations, the SAXS distance distribution can be reproduced for disordered regions. However, the results provided in the manuscript show that often unrestrained CALVADOS-2 simulations have comparable results to CALVADOS-2 simulations with AlphaFold restraints (Figure 2L). This suggests that the AlphaFold restraints do not always provide much benefit over a basic simulation. Thus, the authors need to provide some discussion as to when AlphaFold restraints are helpful versus when the conformational ensemble for disordered proteins is well described by unrestrained simulations. One way the authors could potentially address this is by showing how each model does in reproducing the NMR data that was shown in the previous version of the manuscript. Additionally, the histograms may also provide insights. Overall, this type of discussion must be addressed to solidify the claim that adding AlphaFold restraints are beneficial, at least for certain IDRs.
3. The addition of CALVADOS-2 with only the RMSD restraints for the folded regions curves in Figures 3-6. The authors mention in their response to the reviewers that CALVADOS-2 was not developed to generate structural ensembles. While I agree with this statement, from the methods it appears that they are still using CALVADOS-2 with additional restraints for the folded regions as their structural ensemble for the Metainference simulations. Thus, to actually show that AlphaFold-predicted average distance restraints on disordered regions are beneficial, the control of CALVADOS-2 simulations with the RMSD restraints for the folded regions but without the AlphaFold distance restraints should be provided. Then, if AlphaFold distance restraints on the disordered regions provide helpful predictions on interactions of the disordered regions, the AlphaFold-Metainference simulations should have a lower Kullback-Leibler distance than the simulations without the restraints for the disordered regions. Whatever the result the authors observe, they should provide some explanation for it.

Additional descriptions of these points, as well as additional comments that should be addressed are listed below:

1. The abstract should reflect the application of the method to proteins with both ordered and disordered regions.
2. After introducing the 11 protein set please also add that these IDRs are all relatively expanded with experimentally derived scaling exponents between 0.49 and 0.62. This type of information can better help the reader understand what are the properties of the basis set of IDRs the authors are working with.
3. For Figure 1, insets are provided showing PAE maps and 1D histograms. However, the axes and colorbars used are unreadable. Additionally, there needs to be a description about what a PAE is and what the reader should take away from these plots. Is the error mostly correlated with the distance in sequence space?
4. On page 2, the authors state, "Our results show that there is good agreement between the AlphaFold predictions of distance distribution functions and the SAXS-derived ones." However, this is a vague statement and does not provide the reader with a reference of what would be considered good or bad. The authors should provide a Kullback-Leibler assessment for Figure 1 as well. Additionally, given that the manuscript is trying to show that AlphaFold is comparatively as good for IDRs compared to folded proteins, the authors should show an example folded protein in Figure 1. This will help the reader orient themselves with respect to the error observed in both the distance distributions compared to SAXS, as well as the PAE maps.
5. Please provide a supplemental figure of Figure 1 with the x-axis zoomed to only include up to 21.84 angstroms. Currently it is hard to compare across IDRs because they vary drastically in size. Additionally, please provide the histograms for each of the IDRs in the supplemental. This will provide the reader with information on whether long range interactions are ever predicted by AlphaFold.
6. The authors make the claim that "the AlphaFold distance distribution departs from that obtained by SAXS, because we included in the calculation of the AlphaFold distance distribution all the distances predicted by AlphaFold, independently

from the AlphaFold confidence score.” The authors should further explain what they mean by this. Why should the confidence score matter in this case? Shouldn't the distribution be similar to the SAXS regardless of whether the protein is folded, disordered, or a mixture of both? If the actual confidence score matters, then the authors should provide this information more specifically to the readers. Also RS is not labeled as panel G in Figure 2 even though it is labeled as such in this text.

7. The authors say they are using 11 proteins for which both SAXS and NMR measurements are available. However, in this version the authors do not show any NMR comparisons. Given SAXS and NMR report on different conformational properties, the authors should also add the comparisons to NMR. This data was also shown in the previous version so it is unclear why it is not shown in this version, and the authors do not provide a convincing reason for excluding this data in their responses to the reviewers. Particularly, how well do the three different models do in reproducing the NMR data? Even if part of the focus of this paper ends up now being on the application of the method on partially disordered proteins, this information is still beneficial to have for the IDRs.

8. In general, the paragraph titled “SAXS-based validation of structural ensembles of highly disordered proteins” needs to be expanded on to explain the choice of models, a more detailed description of the goodness of models, some insights as to why models show different results, as well as at least a hypothesis as to why certain proteins show worse similarity to the SAXS data. Also the authors must provide some description in the main text of the “AlphaFold-MetaInference approach” (green curves Figure 2) and how it is different than the model used for the red curves Figure 2. Additionally, the methods section seems to be lacking a section describing how the red curves in Figure 2 were generated. The caption in Figure 2 states, “directly calculated from AlphaFold predictions.”, but this is not explained given the predictions only go up to 22 Å. Please explain how these curves were generated briefly in the main text as well as in the methods. Furthermore, Tau shows the worst agreement with the SAXS data for all the models. Is there some explanation for this? Does the length of the IDR matter?

9. In the methods it is stated, “We choose the PAE cut-off value of 5 Å”. What is the average distance in sequence space that this corresponds to? Are these mostly looking at distances corresponding to residues close in sequence space?

10. What do the circles mean in Figure S1?

11. The authors state in their response to reviewers that, “We thought that we may not report these results in the manuscript since CALVADOS-2 was not intended to be used for partially ordered proteins such as TDP-43.” I am confused by this statement, since the methods suggest that in the AlphaFold-MetaInference method CALVADOS-2 is used to generate the “space of conformations”. Additionally, in the methods it states, “Also, since CALVADOS-2 is a coarse-grained model, we employed the predicted local distance difference test (pLDDT) score in AlphaFold to select the inter-residue distances predicted with higher confidence, which correspond to structured regions. Sequence regions of at least two residues with a pLDDT score >0.75 were hence considered structured regions and restrained to the AlphaFold-predicted structure by using an upper root mean square distance (RMSD) wall.” Thus, just this restraint can be applied to CALVADOS-2 to still provide a comparison of using just restraints on the ordered regions versus AlphaFold restraints on the disordered regions. This control is necessary to provide evidence as to whether the AlphaFold distance restraints on the disordered regions provide improvements compared to unrestrained disordered region simulations.

12. Furthermore, this response does not address the initial concern that raised regarding why are there cases in which CALVADOS-2 unrestrained simulations perform better than AlphaFold restrained simulations, for example for alpha-synuclein? The question initially raised was to try to understand when unrestrained simulations can be used versus when it is necessary to use AlphaFold restraints to get back experimentally consistent conformational ensembles.

13. The authors state in their response to reviewers that “Our main result, however, is now the comparison with the SAXS-derived distance distributions for proteins that contain both ordered and disordered domains.” And they use this reasoning to avoid answering or hypothesizing why some IDR distributions show worse fits to the SAXS data than others. However, the abstract states, “Here we show that the average inter residue distances predicted by AlphaFold for disordered proteins are accurate, and describe how they can be used to construct structural ensembles by incorporating them as structural restraints in molecular dynamics simulations within the metaInference framework. These results illustrate the possibility of making structural predictions for disordered proteins using deep learning methods trained on the large structural databases available for folded proteins.” Thus, the abstract only mentions disordered proteins. Therefore, since disordered regions are still a focus I still raise the question, is this method applicable to all disordered proteins? When is it beneficial to have AlphaFold restraints?

14. While the authors are much clearer in their descriptions of Figure 3-6, they should improve their text describing Figure 2. It would be helpful to more explicitly say that the single structures generated by AlphaFold are not particularly accurate for IDRs (as shown by the red curves in Figure 2). However, when data from the histograms are applied as restraints in simulations they generally are comparable or improve the comparisons to the pairwise distance distributions when compared to unrestrained simulations.

15. Although a comparison to CALVADOS-2 is provided in Figure 2, this is not mentioned in the main text or in the discussion.

16. Add the names of the IDRs as titles to the subplots in Figure 1-2 to make it easier on the reader.

17. The authors state, "AlphaFold-predicted distances as structural restrains in molecular simulations (Figure 2, blue curves)". There are no blue curves in Figure 2 and "restrains" should be "restraints".
18. In the methods it says, "The all-atom ensembles employed in Figures 2, S1, and S2...", but there is no S2.
19. It appears that the color legend in panel A of Figures 3-5 does not match the colors used in the structure shown. Please update this.
20. Please explain the colors used in panel C of Figures 3-5.
21. The legends in Figure 6A-C do not match the curves.
22. The figure caption of figure 6D has many typos.
23. The reference to Figures 3-6 in the discussion should reference Figures 2-6.

Version 2:

Reviewer comments:

Reviewer #2

(Remarks to the Author)

I thank the authors for addressing my previous concerns.

Below are a few minor issues I found in this version of the manuscript.

1. In Figure S6, why are the results for RS, Hst5, and alpha-synuclein not shown? Were distance restraints used for these systems?
2. What do the red and green boxes mean in the Plddt > 75 for residues > 5 residues column in Figure S7A? I assume no vs yes but it would be good to explicitly state this.
3. In the last section of the methods, ataxin-2 is written instead of ataxin-3.

(Remarks on code availability)

Reviewer #1

Brotzakis, Zhang, and Vendruscolo present a very important finding that AlphaFold captures the inter-residues distances of disordered proteins. This is an unexpected finding as AlphaFold was famously trained on folded proteins only. The authors suggest folded and unfolded regions only differ in stability but are otherwise the same and thus AlphaFold is able -in principle - to provide important data on conformational ensembles of disordered regions. They further show how this information can be used to arrive at conformational ensembles of disordered proteins. Rather ingeniously they use the distance map which AlphaFold is using to generate structures to reweight pre-calculated conformational ensembles employing well-established Bayesian/maximum entropy methods. Overall, this manuscript represents an important advance.

We thank the reviewer for the positive assessment of our work.

1) My main suggestion is to provide, e.g., Jupyter notebooks of at least one of the example cases. This will help others to use this approach and develop further improvements which for sure will strengthen the impact of this work further. It will be important to illustrate in such an example how the distance map is extracted from AlphaFold.

We thank the reviewer for this important suggestion. We have now provided all the necessary information to reproduce the results, and corresponding links, in the revised version of the manuscript.

2) In Fig S4-S8 absolute NMR chemical shifts are compared for experiments and computational models. I think it would be better to plot secondary chemical shifts for comparisons so that deviations between models and experiments can be assessed, which is typically what is done for such comparison. In case their CamShift chemical shift prediction method warrants a different approach it would be good to make this clear in the manuscript.

To take into full account point 6 made below, which is indeed crucial to exploit the full potential of our observation that AlphaFold predicts accurately pairwise distances of disordered proteins, we have now implemented the AlphaFold-predicted distances as distance restraints in molecular dynamics simulations. For this reason, we have removed the examples obtained using the reweighting approach, and therefore also the corresponding comparisons with NMR data. We believe that the comparisons with SAXS data for partially disordered proteins shown in the new Figures 3-6 offer now a more compelling and extended validation of the approach than what we provided in the previous version.

3) For α -synuclein an RG value of 36 Å was determined by Araki et al Scientific Reports 2016. This value is often thought as the best experimental estimate as to RG value of α -synuclein. It looks like the reweighed CALVADOS2 ensemble is close to this value. This maybe partially explained by the composition of the CALVADOS2 training set, but it may still be worth mentioning this encouraging agreement with experiments?

The close agreement between our approach and CALVADOS-2 for α -synuclein is now shown in Figure 2K.

4) For the SAXS comparison in Fig 6 it may be good to compare experimental SAXS profiles with a back calculated SAXS profiles at least for one the proteins investigated here. The caption misses an explanation for panel H.

We have now fully revised this figure (now Figure 2 in the revised manuscript) to present the results obtained with the metainference approach.

5) The group of M. Blackledge has studied partially folded α -helices in disordered proteins by NMR. Would the presented approach also be applicable to such cases?

This point is closely related to the following one. Please see our response below.

6) To demonstrate the value of the distance map from AlphaFold, the authors may consider running a restrained (replica averaged) molecular dynamics simulation for a tractable model system. I suspect that such restraints could greatly improve atomistic models of disordered peptides, which could strengthen the key conclusions of this paper.

We are grateful to the reviewer for this very important suggestion. We have now implemented CALVADOS-2 in OpenMM in order to be compatible with PLUMED, and therefore use the AlphaFold distances as restraints in the metainference approach. We now report the results that we obtained for partially disordered proteins (Figures 3-6) to illustrate the potential of the approach.

Minor points

1) For the correlation plots in Figure 2 it would help if the length of the axis is the same (i.e., the plot is square) so that one compare more clearly MD and AlphaFold

Since we now provide a more extensive validation of the ability of AlphaFold of predicting average distances using experimental data (Figure 1), we have removed the comparison with molecular dynamics simulations.

2) Several methods have been proposed to sample different conformational states with AlphaFold. Maybe the authors could comment briefly on whether methods trying to sample conformational diversity may be required for their proposed AlphaFold approach to generate IDP conformational ensembles?

We have now revised the manuscript to better explain that using the distance information as structural restraints within the metainference method is aimed precisely at generating the

maximal possible conformational diversity as allowed by the available information, according to the maximum entropy principle.

3) Could the authors provide some more details as how they energy minimised the models with GROMACS? I.e., which force field was used? I understand that this is a minor point, but including this information may be useful for others who want to build on the presented approach.

For the energy minimization step after building all-atom representations of the structures, we used the Amber99SB-ILDN force field. The complete procedure is provided as a GitHub link in the revised manuscript.

Reviewer #2

In this manuscript Brotzakis et al., compare AlphaFold distance predictions with experimental SAXS distance distributions, as well as show how AlphaFold distance predictions can be utilized to reweight different types of conformational ensembles.

The current version of this manuscript lacks significance to the field, as the authors make general statements that do not reflect the details of the data shown. If the authors can provide additional analyses to show under what conditions AlphaFold predictions are helpful, then the manuscript would be significant to the field.

We are grateful to the reviewer for prompting us to demonstrate more clearly the impact of the observation that we reported in the version of the manuscript that we originally submitted.

To address this essential point, we have implemented our approach in molecular dynamics simulations where the AlphaFold-predicted pairwise distances are used as structural restraints. In this way, we show that it is possible to determine efficiently the conformational ensembles of proteins that contain both ordered and disordered domains. These results are now illustrated in the new Figures 3-6.

More specifically, the authors claim that distances predicted by AlphaFold for disordered proteins are accurate but this statement is too general as the data reported shows caveats to this statement which the authors do not fully address in the text. For example, there are cases where unrestrained coarse-grained models do a better job than coarse-grained models that take into account AlphaFold distance constraints. Thus, it is currently unclear under what conditions one should use AlphaFold constraints versus when these constraints are not helpful. Overall the authors need to provide some additional analyses to show when (which distances and IDR properties) the distances predicted by AlphaFold are accurate as addressed in the points below.

We thank the reviewer for these thoughtful remarks. In Figures 3-6 we now compare the agreement of the pairwise distance distribution probability (PDDF) derived from: (i) SAXS measurements, (ii) AlphaFold-Metainference (AF-MI), (iii) CALVADOS-2, and (iv) AlphaFold (AF) single structures for a series of highly disordered proteins (Figure 2) and partially disordered proteins (Figures 3-6). The comparison is made in terms of a Kullback-Leibler distance (D_{KL}) between the predicted and SAXS-based curve.

Our results indicate that AlphaFold-Metainference decreases the D_{KL} distance with respect to the AlphaFold single-structure calculations in all cases (Figures 2-6). Since CALVADOS-2 was not developed to generate structural ensembles of proteins that contain ordered domains, we have not explicitly carried out CALVADOS-2 simulations for all partially disordered proteins. In one illustrative case of this type, i.e. TDP43-WtoA (Figure R1A, see below), we show here that CALVADOS-2 simulations do not readily generate a structural ensemble in good agreement with SAXS data. For comparison, we also show the case of A1, which is a highly disordered protein, CALVADOS-2 predicts well the SAXS data. We thought that we may not report these results in the manuscript since CALVADOS-2 was not intended to be used for partially ordered proteins such as TDP-43.

Figure R1.1. Comparison of SAXS-derived and AlphaFold-predicted pairwise distance distributions using CALVADOS-2. (A-B) The experimental pairwise distance distributions by SAXS are shown in black, by AlphaFold (AF) in blue, and by the CALVADOS-2 ensemble (C2) in cyan.

Overall, the structural ensembles that we now report for proteins that contain both ordered and disordered domains (Figures 3-6) illustrate that the approach that we describe can offer structural ensembles of better accuracy than otherwise possible with currently available methods.

On page 3, the authors state, “Our results show that there is a good agreement between the AlphaFold predictions of distance distribution functions and the SAXS-derived ones for a set of 7 proteins for which both SAXS measurements NMR diffusion measurements were available.” The authors use 7 of the initial set of 11 IDPs from the paper of Pesce et al. The authors must explain why these 7 were chosen rather than all 11.

We thank the reviewer for making this point. We have now calculated and compared the distance distributions from the AF-MI ensembles with those obtained from SAXS for the entire set of 11 proteins in Pesce et. al. (Figure 2 in the revised manuscript).

Additionally, the authors should provide information about these 7 IDPs including information such as their sequence, sequence length, scaling exponents, etc. This is important since the authors are claiming the AlphaFold prediction shows good agreement but it is not clear whether these IDPs chosen are representative and thus whether this result would hold for any IDP without the additional information about relative representation of the IDPs chosen.

We have now extended the comparison to the whole benchmark set of disordered proteins originally published by Lindorff-Larsen and coworkers (Pesce et al. Biophysical Journal 2023). Our main result, however, is now the comparison with the SAXS-derived distance distributions for proteins that contain both ordered and disordered domains (Figures 3-6).

Particularly, the authors do not compare the AlphaFold predictions of A1 which is the most compact IDP of the Pesce et al., set with the only scaling exponent below 0.5. Thus, it is not clear if A1 was left out because it does not show good agreement or some other reason, and this should be remedied. Overall, unless there is a feasibility issue for the other 4 proteins, the comparison between their SAXS-derived distributions and the AlphaFold predictions should be shown. Even if the authors do not include these 4 IDPs in Figure 6, the additional 4 IDPs need to be included in Figure 1. Without including A1 in particular, it will be unclear whether AlphaFold predictions hold for more compact IDPs.

Additionally, the authors should clarify whether just the disordered regions (i.e., the regions used by Pesce et al.,) or the full protein sequences were used when generating the AlphaFold distograms.

As mentioned above, we have now carried out the calculations for the complete list of 11 proteins in Pesce et. al. in Figure 2.

On page 3, the authors state, "Since AlphaFold predicts distances up to about 22 Å (see Methods), the AlphaFold distribution does not cover the entire SAXS-derived distribution." Since the AlphaFold prediction includes a cutoff it would be helpful to know the distribution of $|j-i|$ lengths that are generally included in these distances, where i and j denote the residue position along the sequence of residue i and j . For instance, are most of the AlphaFold predictions accounting for local conformational information, i.e, within ~ 10 residues along the sequence from each other? How often is long range information ($|j-i| > \sim 20$) observed?

To address this point, we now report the AlphaFold distograms in Figures 3-6, which show the long-range pairwise distances that we used as structural restraints in the molecular dynamics simulations.

Furthermore, according to Wallner et al., (<https://doi.org/10.1101/2022.12.08.519560>) there is relatively good agreement between the histogram distances and the PDB model distances. Thus, it would be helpful to show whether using the PDB model distances using the AlphaFold cutoff of ~ 22 also show good agreement with the SAXS-derived distributions, i.e., how does imposing a single structure with no probability of distances compare to the SAXS-derived distributions? If the PDB model breaks down it would better highlight the importance of using the histogram. If the PDB alone is reasonable, then it would be interesting to know at what length scale the PDB model breaks down since distances greater than ~ 22 can be included.

We now report in Figures 2-6 the comparison of SAXS profiles of the off-the-shelf AlphaFold structures and the structural ensembles generated by our approach.

On page 3, the authors state, "The correlation reported in Figure 2 indicates that AlphaFold can predict inter-residue distances in the disordered native states of Ab and a-synuclein up to about 16 Å, which is about 70% of the cut-off distance in the AlphaFold predictions (see Methods)." How is a cutoff of 16 chosen? A correlation or R^2 should be calculated using different cutoffs to show the validity of this statement.

We have now carried out a sensitivity analysis to optimize the values of the parameters in the simulations (Figure S1), and described our choices in the revised version of the manuscript.

Also what is the average $|j-i|$ that such a cutoff is accounting for? Is this mostly local interactions, i.e., distances between residues close in sequence space. The authors should show if the data is split into distances where $|j-i|$ is less than or equal to ~ 10 and if $|j-i| > \sim 10$ (or a full titration of $|j-i| < x$) whether the distances that correspond to residues close in sequence space have a higher correlation than the distances that correspond to residues farther apart in sequence space. This would also help to know what kind of interactions AlphaFold gets right. Is it just distances between residues close in sequence space or is the variability not restricted to a particular $|j-i|$ range?

We are particularly grateful to the reviewer for this observation, which was for us one of the main motivation of extending the method to proteins containing both ordered and disordered domains. For such proteins, the interactions that are long-range along the chain are essential. The distance maps now shown in Figures 3-6 illustrate this point.

Furthermore, AlphaFold predicted distances are implied to be accurate based on the comparison to the MD extracted distances, yet no comparison of the MD simulations to experimental data is provided at this point in the manuscript (besides a citation) to show that the MD simulations are a good test of accuracy. At least comparing the a-synuclein distance distributions to the SAXS data, since this is being used as the gold standard in this manuscript, should be provided.

We agree that the validation of the AlphaFold-predicted distances by comparison with molecular dynamics simulations (i.e. predictions vs predictions) is not very strong. We have

thus removed this validation. Our central argument for validation is now provided in Figures 2-6 by comparison with SAXS-derived distance distributions.

The representative conformations shown in Figures 3B, 3E, 4B, 5B, and 5F need better contrast since they are hard to see. The authors also need to explain how the representative conformations were picked. It is also not clear what the inclusion of these representative conformations provide.

We have now redrawn all the figures.

On page 5, the authors state, "These results are consistent with previous findings that reweighting with many constraints produces posterior ensembles of high quality even from prior ensembles of relatively quality." There appears to be a word missing before the last "quality".

We have now replaced the reweighting with the more general restraining procedure.

For A β , the C2 and AF-C2 (or just AF) distances are often very different, yet both ensembles show equally good agreement with the NMR data (Figure S6G). The authors should explain what this means, especially in the context of the fact that there are no experimental distances to compare to so which model is correct is unclear. Thus, if both do an equally good agreement with the NMR data is there a time when people should just use C2 vs AF-C2? Also in the main text the authors refer to Figure S6F when they seem to mean S6G.

On page 7, the authors state, "The AF-MD ensemble shows marginal improvement with respect to the MD ensemble in the chemical shifts of all types of atoms (Figure S7F)." It is unclear why the authors claim "marginal improvement...in the chemical shifts of all types of atoms" when only cb and maybe hn show a lower RMSD in AF-MD compared to MD. Thus, this statement needs to be updated to reflect the data.

We have now focused the validation on the challenging case of proteins that contain both ordered and disordered regions, for which we used SAXS data for validation (Figures 3-6).

On page 8, the authors state, "The use of the AlphaFold-predicted pairwise distances as constraints in the reweighting procedure tended to slightly improve the agreement with the experimental data, as measured using the Kullback-Leibler distance (see Methods) between distributions (Figure 6K)". First, the authors appear to be referring to 6H not 6K. Second, only ProTa and GHR-ICD have smaller Kullback-Leibler distance given the error bars, thus it is unclear why the authors claim AlphaFold-predicted constraints generally improve agreement. Thus, this statement is incorrect and must be changed to reflect the data. Third, the figure caption of Figure 6 does not match the subplots. Although it could be assumed panels A-G refer to the proteins in the order they are listed in H this should be clarified on the plot or in the figure caption. Additionally, the authors must also clarify what the error bars mean in panel H.

We have now replaced these results with those obtained using metainference.

Lastly, the authors spend a decent amount of the paper showing how α -synuclein ensembles can be reweighted to match AlphaFold-predicted distances only to show that the C2 model without reweights is a better approximation of the experimental data. Thus, it is unclear when / why reweighting to AlphaFold-predicted distances would be helpful. Given that some IDPs do better using the AlphaFold-predicted constraints and some IDPs do worse / the same, the authors should provide some sort of analysis to try to explain this and thus give the readers a sense of when using the AlphaFold-predicted constraints would be more beneficial than just using the C2 model.

We have now reported results for partially disorderd proteins, where the improvement with respect to CALVADOS-2 is always present (Figure R1 and Figure 3D in the manuscript).

Figure R2.1. Comparison of SAXS-derived and AlphaFold-predicted pairwise distance distributions using CALVADOS-2. (A-B) The experimental pairwise distance distributions by SAXS are shown in black, by AlphaFold (AF) in blue, and by the CALVADOS-2 ensemble (C2) in cyan.

Additionally, again the authors should explain why only 7 of the 11 Pesce et al., proteins were chosen to examine in Figure 6. Unless there is a feasibility issue, the remaining 4 proteins should also be included here.

We have now reported all the 11 proteins.

In the abstract, the authors statement “inter-residue distances predicted by AlphaFold for disordered proteins are accurate”. This statement needs to be toned down and caveats need to be included.

We have now better explained this statement.

On page 10, the authors state, “In practice, rather than constraining all distances, we did not include neighbouring residues (up to 2 residues) for $A\beta$ and that have a distance larger than 21.8 Å in the MD or FD ensembles. For the α -synuclein calculations, we did not include

neighbouring residues (up to 2 residues) and that have a distance larger than 21.8 Å in the MD ensemble.” Why are these constraints only used for some of the ensembles? i.e., were these same constraints not used for the C2 ensembles?

We have now specified that we have included these constraints.

A general issue that must be addressed is the use of comparison to SAXS derived distance distributions to show that AlphaFold is accurate (Figure 1). Given the data shown in Figure 6, it appears that a coarse-grained model with no constraints is often comparable to the AlphaFold distance probabilities up to ~20 Å. Thus, the question becomes are all models equally good at comparing to SAXS derived distances distributions up to this cutoff? If all models are equally good, then how does that imply AlphaFold is accurate / why should one use AlphaFold distances? I suggest that the authors provide a Kullback-Leibler distance measure for the single structure PDB from AlphaFold, the AlphaFold histogram distances, and the C2 model up to 22 Å. At least some comparison of accuracy should be included since the authors want to make the claim that using AlphaFold distances are helpful.

We agree with the reviewer that this is an important point. In order to address it as exhaustively as possible, we have now implemented the method in metainference and applied it to proteins that contain both ordered and disordered regions. Our results show that structural ensembles for these proteins cannot be accurately obtained by the off-the-shelf AlphaFold or by methods designed specifically for disordered proteins, such as CALVADOS-2.

On page 8, the authors state, “Overall, the results that we have presented illustrate the use of deep learning methods originally developed for predicting the native states of folded proteins 1-3 to generate structural ensembles representing the native states of disordered proteins, or of proteins containing disordered regions.” This statement is too general and must be changed to reflect what the data shows. The AlphaFold predictions are not always beneficial. Thus, the authors need to provide analyses to show when the AlphaFold predictions are beneficial to avoid misleading claims.

We have now revised this statement to take into account both the comments by the reviewer and the new results obtained using the metainference approach.

Reviewer #2

Overall the new results and text provided for the partially disordered proteins is clear. However, the authors should button up the text on the disordered proteins and show a few more analyses / controls in order to clean up the full story. The general takeaway the authors appear to describe is that using the AlphaFold structure directly from the web server is not particularly good at predicting the conformational ensemble. However, when AlphaFold-predicted average distances are used as restraints for disordered regions in simulations, then conformational ensemble information is in good agreement to SAXS.

We thank the reviewer for capturing the essence of our results, and for providing further useful suggestions for improving our presentation of the AlphaFold-Metainference method.

To support this takeaway and the need for using AlphaFold restraints the authors should at least provide the following:

1. Distograms predicted by AlphaFold for each of the disordered proteins in the supplementary. These plots will show whether AlphaFold predicted distances ever show long range interactions or whether the restraints are generally just on local interactions.

We have now added the AlphaFold2 distograms of the proteins considered (Figure S3), as well as their corresponding predicted alignment error (PAE) maps (Figure S8). Distances with low PAE values are characterised by both low errors and high accuracy in the predictions. By contrast, distances with high PAE values can either report on flexibility of interactions or uncertainty in the prediction. To take into account these multiple possibilities, we used metainference, which is a method developed to disentangle the errors in data from structural heterogeneity.

This could also be provided by supplementary figures that show the average sequence separation or distribution of sequence separations used for the restraints. Additionally, the authors need to say whether these restraints are mainly on local interactions or not in the main text.

To take this helpful suggestion into account, we included an additional figure (Figure S6) showing the sequence separation of the pairwise distances predicted by AlphaFold that abide to the distance selection rules and are used as structural restraints in the AlphaFold-Metainference simulations. We comment on this figure in the main text in the sections "*SAXS-based validation of structural ensembles of highly disordered proteins*" and "*SAXS-based validation of structural ensembles of partially disordered proteins*".

2. A more detailed description of Figure 2 in the main text (see point 8 below). Currently the main text does not mention what the models for each of the curves are. This description should include a comparison between CALVADOS-2 results with and without restraints. One of the paper's main claims is that when AlphaFold-predicted average distances are used as restraints in simulations, the SAXS distance distribution can be reproduced for disordered regions. However, the results provided in the manuscript show that often unrestrained CALVADOS-2 simulations have comparable results to CALVADOS-2 simulations with AlphaFold restraints (Figure 2L). This suggests that the AlphaFold restraints do not always provide much benefit over a basic simulation. Thus, the authors need to provide some discussion as to when AlphaFold restraints are helpful versus when the conformational ensemble for disordered proteins is well described by unrestrained simulations. One way the authors could potentially address this is by showing how each model does in reproducing the NMR data that was shown in the previous version of the manuscript. Additionally, the distograms may also provide insights. Overall, this type of discussion must be addressed to solidify the claim that adding AlphaFold restraints are beneficial, at least for certain IDRs.

We thank the reviewer for prompting us to address this important comparison (see also point 7 below). In addition to the comparison described at point 1 above, we have now performed a more extensive analysis and presented it in the section "*SAXS-based validation of structural ensembles of highly disordered proteins*", including Figures S4-S6. We also note that the main advantage of using AlphaFold-Metainference could be expected in situations where proteins contain both ordered and disordered regions, see section "*SAXS-based validation of structural ensembles of partially disordered proteins*".

3. The addition of CALVADOS-2 with only the RMSD restraints for the folded regions curves in Figures 3-6. The authors mention in their response to the reviewers that CALVADOS-2 was not developed to generate structural ensembles. While I agree with this statement, from the methods it appears that they are still using CALVADOS-2 with additional restraints for the folded regions as their structural ensemble for the Metainference simulations. Thus, to actually show that AlphaFold-predicted average distance restraints on disordered regions are beneficial, the control of CALVADOS-2 simulations with the RMSD restraints for the folded regions but without the AlphaFold distance restraints should be provided. Then, if AlphaFold distance restraints on the disordered regions provide helpful predictions on interactions of the disordered regions, the AlphaFold-Metainference simulations should have a lower Kullback-Leibler distance than the simulations without the restraints for the disordered regions. Whatever the result the authors observe, they should provide some explanation for it.

This analysis has now been added in Figure 6D. For most of the cases, the AlphaFold-Metainference simulations improve over the simulations with CALVADOS-2 with RMSD restraints on the ordered region (abbreviated RMSD-C2), especially for proteins with many AlphaFold-predicted long-range distances (Figure S6). We have now added a discussion on this point in the section "*SAXS-based validation of structural ensembles of partially disordered proteins*".

Additional descriptions of these points, as well as additional comments that should be addressed are listed below:

1. The abstract should reflect the application of the method to proteins with both ordered and disordered regions.

We have now modified the Abstract to reflect the changes made for the resubmission.

2. After introducing the 11 protein set please also add that these IDRs are all relatively expanded with experimentally derived scaling exponents between 0.49 and 0.62. This type of information can better help the reader understand what are the properties of the basis set of IDRs the authors are working with.

These values are now reported in Figure S4. We have clarified it in the section "*SAXS-based validation of structural ensembles of highly disordered proteins*".

3. For Figure 1, insets are provided showing PAE maps and 1D histograms. However, the axes and colorbars used are unreadable. Additionally, there needs to be a description about what a PAE is and what the reader should take away from these plots. Is the error mostly correlated with the distance in sequence space?

We have now reported the PAE maps in Figure S8, and introduced the PAE score in the Methods section.

4. On page 2, the authors state, "Our results show that there is good agreement between the AlphaFold predictions of distance distribution functions and the SAXS-derived ones." However, this is a vague statement and does not provide the reader with a reference of what would be considered good or bad. The authors should provide a Kullback-Leibler assessment for Figure 1 as well. Additionally, given that the manuscript is trying to show that AlphaFold is comparatively as good for IDRs compared to folded proteins, the authors should show an example folded protein in Figure 1. This will help the reader orient themselves with respect to the error observed in both the distance distributions compared to SAXS, as well as the PAE maps.

We now report the Kullback-Leibler values assessment in the amended Figure 1. We have also provided a control comparison between the AlphaFold-predicted distance distribution function and SAXS for ubiquitin. Moreover, we have added the respective distogram and PAE map in Figures S3

and S8. We find a comparable D_{KL} (0.037) for ubiquitin with respect to the 11 IDPs (D_{KL} range: 0.008-0.096).

5. Please provide a supplemental figure of Figure 1 with the x-axis zoomed to only include up to 21.84 angstroms. Currently it is hard to compare across IDRs because they vary drastically in size. Additionally, please provide the histograms for each of the IDRs in the supplemental. This will provide the reader with information on whether long range interactions are ever predicted by AlphaFold.

We have now provided with a zoomed version of Figure 1 in Figure S2. The histogram for each IDP is reported in Figure S3.

6. The authors make the claim that "the AlphaFold distance distribution departs from that obtained by SAXS, because we included in the calculation of the AlphaFold distance distribution all the distances predicted by AlphaFold, independently from the AlphaFold confidence score." The authors should further explain what they mean by this. Why should the confidence score matter in this case? Shouldn't the distribution be similar to the SAXS regardless of whether the protein is folded, disordered, or a mixture of both? If the actual confidence score matters, then the authors should provide this information more specifically to the readers

AlphaFold-predicted distances do indeed provide structural information both for disordered and ordered proteins. Yet, as illustrated in Figure S8, these distances are characterised by different PAE values. This prediction error can be attributed either to the intrinsic dynamics or the error in the AlphaFold predictions. Hence, not all AlphaFold distances are equally accurate, thus contributing to the discrepancy to SAXS data when we plot all AlphaFold distances in a pair distance distribution function in Figure 1. To explain more accurately this point, we revised this section in the manuscript.

Also RS is not labeled as panel G in Figure 2 even though it is labeled as such in this text.

We have fixed this typo.

7. The authors say they are using 11 proteins for which both SAXS and NMR measurements are available. However, in this version the authors do not show any NMR comparisons. Given SAXS and NMR report on different conformational properties, the authors should also add the comparisons to NMR. This data was also shown in the previous version so it is unclear why it is not shown in this version, and the authors do not provide a convincing reason for excluding this data in their responses to the reviewers. Particularly, how well do the three different models do in reproducing the NMR data? Even if part of the focus of this paper ends up now being on the application of the method on partially disordered proteins, this information is still beneficial to have for the IDRs.

The previous comparison with NMR concerned α -synuclein and A β , whose structural ensembles were obtained by using a reweighting approach. During the first round of revisions, we introduced instead a restraining approach - the AlphaFold-Metainference method (AF-MI) - which incorporates AlphaFold-predicted distances as structural restraints using metainference. This approach is more efficient than reweighting to generate structural ensembles that match experimental data according to the maximum entropy principle. Hence, in AF-MI we focused on a wider set of proteins for which there was richer information in terms of abundant SAXS and some NMR data. For these proteins, we found available NMR data for Sic1 and AN16 and presented the resulting validation in Figure S5.

8. In general, the paragraph titled "SAXS-based validation of structural ensembles of highly disordered proteins" needs to be expanded on to explain the choice of models, a more detailed description of the goodness of models, some insights as to why models show different results, as well as at least a hypothesis as to why certain proteins show worse similarity to the SAXS data.

We have now extended this paragraph following the discussion of point 2 above.

Also the authors must provide some description in the main text of the "AlphaFold-MetaInference approach" (green curves Figure 2) and how it is different than the model used for the red curves Figure 2.

We have described this method in more detail in the revised version of the manuscript.

Additionally, the methods section seems to be lacking a section describing how the red curves in Figure 2 were generated. The caption in Figure 2 states, "directly calculated from AlphaFold predictions.", but this is not explained given the predictions only go up to 22 Å. Please explain how these curves were generated briefly in the main text as well as in the methods. Furthermore, Tau shows the worst agreement with the SAXS data for all the models. Is there some explanation for this? Does the length of the IDR matter?

We have now explained that the single structures were generated using the AlphaFold Protein Structure Database and not by AlphaFold-MetaInference.

9. In the methods it is stated, "We choose the PAE cut-off value of 5 Å". What is the average distance in sequence space that this corresponds to? Are these mostly looking at distances corresponding to residues close in sequence space?

We now show the sequence separation of the distances in Figure S6, using the criterion described in point 2 and in Figure S7.

10. What do the circles mean in Figure S1?

We have now replaced this figure and the corresponding discussion to explain the distance selection criterion in Figure S7.

11. The authors state in their response to reviewers that, "We thought that we may not report these results in the manuscript since CALVADOS-2 was not intended to be used for partially ordered proteins such as TDP-43." I am confused by this statement, since the methods suggest that in the AlphaFold-MetaInference method CALVADOS-2 is used to generate the "space of conformations". Additionally, in the methods it states, "Also, since CALVADOS-2 is a coarse-grained model, we employed the predicted local distance difference test (pLDDT) score in AlphaFold to select the inter-residue distances predicted with higher confidence, which correspond to structured regions. Sequence regions of at least two residues with a pLDDT score >0.75 were hence considered structured regions and restrained to the AlphaFold-predicted structure by using an upper root mean square distance (RMSD) wall." Thus, just this restraint can be applied to CALVADOS-2 to still provide a comparison of using just restraints on the ordered regions versus AlphaFold restraints on the disordered regions. This control is necessary to provide evidence as to whether the AlphaFold distance restraints on the disordered regions provide improvements compared to unrestrained disordered region simulations.

We have now performed this comparison and reported in Figure 6L, as discussed this at point 3.

12. Furthermore, this response does not address the initial concern that raised regarding why are there cases in which CALVADOS-2 unrestrained simulations perform better than AlphaFold restrained simulations, for example for alpha-synuclein? The question initially raised was to try to understand when unrestrained simulations can be used versus when it is necessary to use AlphaFold restraints to get back experimentally consistent conformational ensembles.

We agree with the referee that this is an important point. We have now addressed it in the revised version of the manuscript, see in particular Figures 2 and 6.

13. The authors state in their response to reviewers that "Our main result, however, is now the comparison with the SAXS-derived distance distributions for proteins that contain both ordered and

disordered domains.” And they use this reasoning to avoid answering or hypothesizing why some IDR distributions show worse fits to the SAXS data than others. However, the abstract states, “Here we show that the average inter residue distances predicted by AlphaFold for disordered proteins are accurate, and describe how they can be used to construct structural ensembles by incorporating them as structural restraints in molecular dynamics simulations within the metainference framework. These results illustrate the possibility of making structural predictions for disordered proteins using deep learning methods trained on the large structural databases available for folded proteins.” Thus, the abstract only mentions disordered proteins. Therefore, since disordered regions are still a focus I still raise the question, is this method applicable to all disordered proteins? When is it beneficial to have AlphaFold restraints?

We thank the referee for prompting us to clarify this point. We have now addressed it in the revised version of the manuscript. The AlphaFold-Metainference method is beneficial when at least some AlphaFold-predicted distances are of sufficiently high reliability, according to the pLDDT values.

14. While the authors are much clearer in their descriptions of Figure 3-6, they should improve their text describing Figure 2. It would be helpful to more explicitly say that the single structures generated by AlphaFold are not particularly accurate for IDRs (as shown by the red curves in Figure 2). However, when data from the distograms are applied as restraints in simulations they generally are comparable or improve the comparisons to the pairwise distance distributions when compared to unrestrained simulations.

In the revised manuscript, we have now clarified the text describing Figure 2.

15. Although a comparison to CALVADOS-2 is provided in Figure 2, this is not mentioned in the main text or in the discussion.

We have now discussed in more detail the results in Figure 2 (see point 8 above), in particular concerning CALVADOS-2.

16. Add the names of the IDRs as titles to the subplots in Figure 1-2 to make it easier on the reader.

We have addressed this point by amending Figures 1 and 2.

17. The authors state, “AlphaFold-predicted distances as structural restrains in molecular simulations (Figure 2, blue curves)”. There are no blue curves in Figure 2 and “restrains” should be “restraints”.

We have addressed this point.

18. In the methods it says, “The all-atom ensembles employed in Figures 2, S1, and S2...”, but there is no S2.

Figure S2 is in the revised Supporting Information.

19. It appears that the color legend in panel A of Figures 3-5 does not match the colors used in the structure shown. Please update this.

We have now updated the color legends and reported for Figures 3-5.

20. Please explain the colors used in panel C of Figures 3-5.

In this panel the colors correspond to different secondary structure according to Chimera visualization tool - cyan corresponds to loop, blue to helices and red to b-sheet. A color panel has been added.

21. The legends in Figure 6A-C do not match the curves.

This error has been fixed.

22. The figure caption of figure 6D has many typos.

This error has been fixed.

23. The reference to Figures 3-6 in the discussion should reference Figures 2-6.

These 3 points are now amended.

Reviewer #2's

I thank the authors for addressing my previous concerns.

Below are a few minor issues I found in this version of the manuscript.

1. In Figure S6, why are the results for RS, Hst5, and alpha-synuclein not shown? Were distance restraints used for these systems?

We have now explained more clearly that none of the inter-residue distances predicted by AlphaFold for these 3 protein met the conditions for being included in the simulations.

2. What do the red and green boxes mean in the Plddt > 75 for residues > 5 residues column in Figure S7A? I assume no vs yes but it would be good to explicitly state this.

We have provided this information.

3. In the last section of the methods, ataxin-2 is written instead of ataxin-3.

We have corrected this typo.